# Litterfall Production and Litter Decomposition Experiments: In Situ Datasets of Nutrient Fluxes in Two Bornean Lowland Rain Forests Associated with *Acacia* Invasion

Salwana Md. Jaafar [1,*] , Rahayu Sukmaria Sukri [1] , Faizah Metali [2] and David F. R. P. Burslem [3]

[1] Institute for Biodiversity and Environmental Research, Universiti Brunei Darussalam, Bandar Seri Begawan BE 1410, Brunei
[2] Environmental and Life Sciences Programme, Faculty of Science, Universiti Brunei Darussalam, Bandar Seri Begawan BE 1410, Brunei
[3] School of Biological Sciences, University of Aberdeen Cruickshank Building, Aberdeen AB24 3UL, UK
* Correspondence: salwana.jaafar@ubd.edu.bn

**Abstract:** AbstractIt is increasingly recognized that invasion by alien plant species such as *Acacia* spp. can impact tropical forest ecosystems, although quantifications of nutrient fluxes for invaded lowland tropical rain forests in aseasonal climates remain understudied. This paper describes the methodology and presents data collected during a year-long study of litterfall production and leaf litter decomposition rates in two distinct tropical lowland forests in Borneo affected by *Acacia* invasion. The study is the first to present a comprehensive dataset on the impacts of invasive *Acacia* species on Bornean forests and can be further used for future research to assess the long-term impact of *Acacia* invasion in these forest ecosystems. Extensive studies of nutrient cycling processes in aseasonal tropical lowland rainforests occurring on different soil types remain limited. Therefore, this dataset improves understanding of nutrient cycling and ecosystem processes in tropical forests and can be utilized by the wider scientific community to examine ecosystem responses in tropical forests.

**Keywords:** alien plant species; aseasonal climate; biogeochemical cycling; Brunei; heath forests; Kerangas forests; mixed dipterocarp forests; nutrient cycling

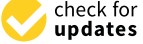



## 1. Summary

Nutrient fluxes from litterfall production and litter decomposition are important processes in biogeochemical cycling in forest ecosystems and may reflect production dynamics and nutrient balances [1,2]. Under aseasonal climate conditions and for lowland forests in Southeast Asia, quantitative and qualitative data on nutrient fluxes remain limited [3] as compared to data gathered from seasonal climate conditions in Central Africa and South America [2]. Additionally, nutrient fluxes in forest ecosystems can vary depending on climate, tree species composition and forest structure [4]. Invasion by alien plant species are known to impact many aspects of nutrient flux in an ecosystem. For example, invasive plants can potentially increase soil nutrient content and plant biomass in invaded ecosystems [5]. Rapid growth of invasive plant species typically results in high litterfall production [6] and produces nutrient-rich litter that decompose faster [7], hence disrupting nutrient transfer processes in the invaded forest ecosystem to the advantage of the invasive species.

*Acacia* has been recognized as one of the most invasive genera of plant species [8] in tropical countries in East Asia and Southeast Asia [9]. In invaded environments, *Acacia*

species produce high total litterfall and nutrient-rich litter with high nitrogen (N) concentrations [10,11]. The high production of nutrient-rich litter increases the available nutrients in the soil and the ecosystem [10–12] which will eventually impact native plant species that are often adapted to low or moderate nutrient environment. In aseasonal Northwest Borneo, specifically in Brunei Darussalam, *Acacia* was introduced in the 1990s for soil mitigation and plantation forestry [13]. This invasive species has now dominated the landscapes of disturbed and secondary forests in Brunei Darussalam [14,15] and has similarly invaded two lowland forest types: heath forest (HF) and mixed dipterocarp forest (MDF) [16].

The impact of invasive *Acacia* species on nutrient fluxes in Bornean lowland tropical forests remains little studied. This paper provides a set of data and describes the methodology for the quantification of two major components of nutrient flux, litterfall production and leaf litter decomposition rates, from HF and MDF sites in Brunei Darussalam (see Jaafar et al. [17]). Litterfall production was quantified via measurements of litterfall production over a one-year study period, as well as variation in nutrient concentrations, nutrient addition, and stand-level nutrient efficiency (NUE) of leaf litterfall production, whereas leaf litter decomposition was measured in terms of litter decomposition rates, variation in litter pH and nutrient remaining. The nutrients measured include the concentration of nitrogen (N), phosphorus (P), potassium (K), calcium (Ca) and magnesium (Mg). These datasets represent the first known study published on the impact of *Acacia* invasion on Borneo's high-conservation-value forests and are invaluable as a baseline to assess the long-term impact of *Acacia* invasion in these aseasonal lowland tropical rainforests. Publicly accessible and comprehensive datasets on the nutrient cycling of these forests are still limited, and thus our dataset is instrumental to future research on the nutrient cycling processes of intact tropical forests. Broadly, the dataset is valuable for the wider scientific community and can contribute towards increased understanding of the impacts of invasive species on forest ecosystems and management of native biodiversity in the presence of exotic species, and for improved modelling of ecosystem responses to plant invaders and the interactions of plant invasions with climate change. Practically, the dataset can be utilized by forest managers and local policy makers to formulate mitigation and control measures addressing the introduction of new species.

## 2. Data Description

The study was designed to examine the effects of *A. mangium* on variation in total litterfall production and leaf litter decomposition at the two forest types and hypothesized that *Acacia* would impact measures of nutrient fluxes differently in distinct forest types. The data collection was conducted over a 12-month period of continuous monitoring of litterfall production, measurements of leaf litterfall nutrient concentrations, nutrient addition via leaf litterfall, stand-level nutrient use efficiency (NUE), percentage of litter mass remaining and nutrient remaining via leaf litter decomposition (Figure 1, Table 1). Data was acquired via field sampling, measurement, and chemical analysis and the formats given are raw and analyzed data. Field data were collected in 2016–2017 using standard protocols found in "Litterfall Monitoring Protocol CTFS Global Forest Carbon Research Initiative" by Mueller-Landau & Wright [18] and following Dutta et al. [19] and Cizungu et al. [20] for the litter decomposition rates experiment.

Interpretation of data analyses have been published in Jaafar et al. [17]. Overall, the results show that *Acacia*-invaded forests exhibited higher total litterfall production, increased leaf litter concentrations of N, K and Ca, and increased addition of all nutrients measured in litter (N, P, K, Ca, and Mg, especially in the *Acacia*-invaded mixed dipterocarp forest (AMDF) and N and K in *Acacia*-invaded heath forest (AHF)), reduced N and K use efficiencies in AHF, and reduced stand-level N and Ca use efficiencies in AMDF. Litter decomposition rates and nutrient release were lower in AMDF than in the three other habitats. The significantly higher total litterfall production coupled with higher nutrient addition in the two *Acacia*-invaded habitats is expected to progressively increase the abilities

of these habitats to produce large quantities of nutrient-rich litter and will likely eventually lead to an enrichment of nutrients in the soil, thus facilitating further invasion by *Acacia*.

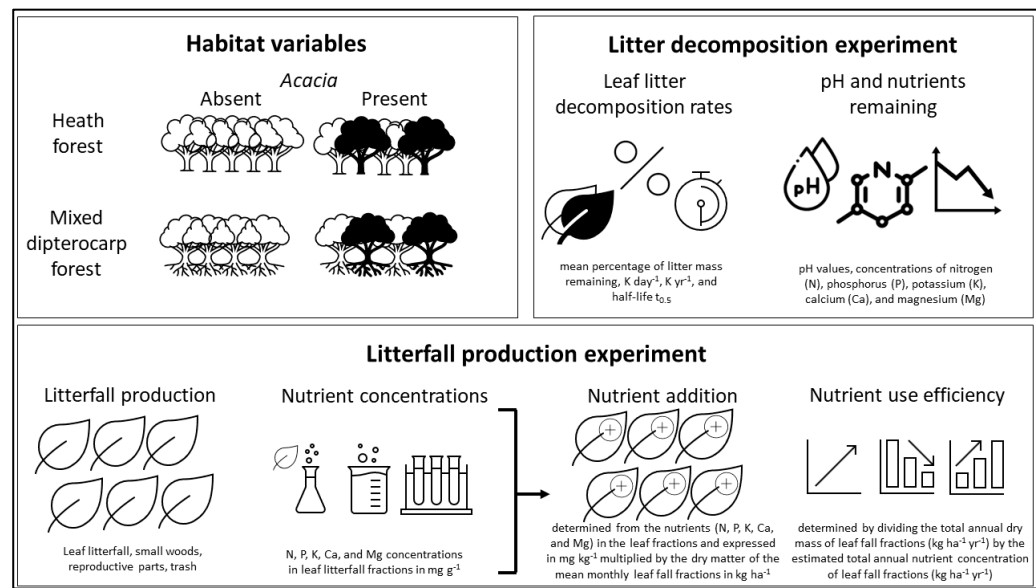

**Figure 1.** Graphical overview of the habitat variables and parameters included in the litterfall production and litter decomposition experiments.

**Table 1.** Summary of the type and volume of data provided. This data article contains seven excel files.

| | GPS | Litterfall Production | Nutrient Concentration | Nutrient Addition | Nutrient Use Efficiency | Leaf Litter Decomposition Rates | Litter pH & Nutrient Remaining |
|---|---|---|---|---|---|---|---|
| Number of points | 24 | 96 | 96 | 96 | 96 | 864 | 864 |
| Number of samples | - | 1140 | 281 | 48 | 48 | 46 | 46 |
| Number of parameters | - | 4 | 6 | 5 | 5 | 4 | 6 |
| Sampling period | February 2016 | October 2016–September 2017 | January 2017–June 2018 | January 2017–June 2018 | January 2017–June 2018 | January 2017–June 2018 | January 2017–June 2018 |

## 3. Methods

### 3.1. Study Sites

Brunei Darussalam is located in Northwest Borneo surrounded by Sabah and Sarawak, Malaysia and Kalimantan, Indonesia. The study site took place in two aseasonal lowland forest types, heath forest (HF) and mixed dipterocarp forest (MDF) covering approximately 8 ha of the forest areas of Andulau Forest Reserve in the Belait District, Brunei Darussalam (4°37′24.58″ N 114°32′53.46″ E). The two forest types were further divided into four habitat types as follows: intact heath forest (HF), *Acacia*-invaded heath forest (AHF), intact mixed dipterocarp forest (MDF), and *Acacia*-invaded mixed dipterocarp forest (AMDF). In February 2016, a total of twenty-four 20 m × 20 m plots were randomly established within accessible locations across all four habitat types (six plots per habitat type). Each 20 × 20 m plot was further subdivided into four 10 × 10 m subplots.

### 3.2. Litterfall Production Data Collection

In August 2016, one above-ground litterfall trap was placed in each of the 10 m × 10 m subplots, for a total of 96 aboveground litterfall traps in all 24 plots. Litterfall trap construction followed Mueller-Landau & Wright [18]. Litterfall traps were placed

randomly in the 10 m × 10 m subplots, avoiding gaps in the canopy. Litter from the litterfall traps was collected once a month from October 2016 to September 2017 to cover a 12-month period. Collected litter samples were oven dried at 60 °C for 48 h and sorted into the following fractions according to Proctor et al. [21]: Leaf litter, small woody debris (fragments approximately 2 mm to 50 cm in size), reproductive parts (fruit and flower parts), and trash (all parts that fit through a 1 cm mesh sieve, including dead invertebrates and feces). Each fraction was weighed to the nearest 0.1 g for each litterfall trap and collection date, and total monthly litterfall production was calculated by summing the four fractions.

### 3.3. Leaf Litter Decomposition Rates Data Collection

The litter bag technique was used to quantify decomposition rates of leaf litter [19,20]. Litter bags (20 × 20 cm) of 2 mm nylon fabric were filled with approximately 20 g of air-dried leaf litter that had been thoroughly mixed. Two types of leaf litter were used: (1) *Acacia* litter as standard litter samples and (2) mixed litter—a mixture of leaf litter from native (and invasive) tree species present in the study plots. Only freshly fallen leaf litter with no signs of damage was collected over two months (July–August 2016).

A total of 864 litter bags were prepared for the leaf litter decomposition experiment. In each 20 m × 20 m plot, two parallel transect lines 10 m apart and approximately 5 m long were randomly established and marked with colored tubes to indicate the locations of *Acacia* litter bags and mixed species litter bags. The bags were randomly distributed along the 48 transect lines. Along each transect line, 18 litter bags were buried 0.5 m apart under forest floor litter (approximately 3 cm deep) to maximize contact with decomposer organisms. The litter bags were disposed of in October 2016, and the first and last litter bag collections were conducted in October 2016 and September 2017, respectively.

Three litter bags per litter type or transect line were randomly collected from each of 24 plots in four habitats at 14, 21, 42, 84, 168, and 336 days. Litter samples collected from each litter bag were cleaned of debris such as soil, roots, and small insects. The cleaned samples were oven dried at 60 °C for 72 h, and the dry mass of the remaining litter samples in each litter bag was determined.

### 3.4. Chemical and Data Analyses Procedures

Only leaf litter fractions were used to determine nutrient concentrations. For each monthly collection, leaf fractions (approximately 20 g) were randomly selected from one litterfall trap per plot (20 m × 20 m). For the leaf litter decomposition experiment, the three litter bags per litter type collected in one day were combined and treated as one sample. Samples from both studies were analyzed separately for pH, N, P, K, Ca, and Mg concentrations. Before all chemical analyses were performed, the oven-dried samples were reduced in size by crushing and grinding with a ball mill (Retch GmbH Mixer Mill MM400, Germany). The pH was measured following Perez-Harguindeguy et al. [22], while the N, P, K, Ca, and Mg concentrations were determined using a modified procedure described by Allen et al. [23] for N and P using the Kjeldahl method and measured using a Flow Injector Analyzer (FIAstar 5000, Hoganas, Sweden), and for Mg, Ca and K measured using a Flame Atomic Absorption Spectrophotometer (AAS; Thermo Scientific iCE 3300, Sydney, Australia). The chemical analysis procedures are described in detail in Jaafar et al. [17].

To calculate total monthly litterfall production in a 1 ha area, dry masses of mean total litterfall and its fractions per habitat type were converted to kg ha$^{-1}$. Based on the dry masses recorded for the monthly leaf fractions and the nutrient concentrations obtained from subsamples of the leaf litterfall fractions, the estimated nutrient addition from leaf litterfall and the nutrient use efficiency (NUE) index were calculated. The estimated nutrient addition from leaf production at each study site for one year (October 2016–September 2017) was calculated following Dent et al. [24] from nutrients (N, P, K, Ca, and Mg) determined in leaf fractions and expressed in mg kg$^{-1}$ multiplied by dry masses of mean monthly leaf fall fractions in kg ha$^{-1}$. Estimated stand-level NUE was determined by dividing the

total annual dry mass of leaf litter fractions (kg ha$^{-1}$ yr$^{-1}$) by the estimated total annual nutrient concentration of leaf fractions (kg ha$^{-1}$ yr$^{-1}$), following Vitousek et al. [25,26] and Moran et al. [27].

For the leaf litter decomposition experiment, the values for the percentage of litter mass remaining, decay coefficients ($K$ day$^{-1}$ and $K$ yr$^{-1}$), half-lives ($t_{0.5}$ in days), and percentages of nutrients remaining were determined according to the formulas described in Dent et al. [24], Sukri [28], and Suhaili [29]. All parameters were calculated separately for *Acacia* and mixed litter for all 24 plots after 14, 21, 42, 84, 168, and 336 days of the leaf litter decomposition experiment. Calculations and formulas are described in detail in Jaafar et al. [17].

**Author Contributions:** Conceptualization, R.S.S., S.M.J., F.M. and D.F.R.P.B.; methodology, S.M.J., R.S.S. and F.M.; formal analysis, S.M.J. and R.S.S.; investigation, S.M.J.; resources, R.S.S., F.M. and D.F.R.P.B.; data curation, S.M.J.; writing—original draft preparation, S.M.J. and R.S.S.; writing—review and editing, S.M.J., R.S.S., F.M. and D.F.R.P.B.; supervision, R.S.S. and F.M.; project administration, R.S.S.; funding acquisition, R.S.S. All authors have read and agreed to the published version of the manuscript.

**Funding:** The research was funded by the Brunei Research Council [Grant No. UBD/BRC/11] and Universiti Brunei Darussalam [Grant Nos. UBD/RSCH/1.4/FICBF(b)/2018/005 and UBD/PNC2/2/RG/1(204)]. SMJ was supported by a University Graduate Scholarship (UGS) from Universiti Brunei Darussalam.

**Institutional Review Board Statement:** Not applicable.

**Informed Consent Statement:** Not applicable.

**Data Availability Statement:** The data presented in this study are openly available in the Zenodo repository at https://doi.org/10.5281/zenodo.7528856 (accessed on 12 January 2023).

**Acknowledgments:** The authors thank the Brunei Forestry Department for the use and collection permit (Permit No. [99]/JPH/UND/17 PT.1), the Department of Agriculture and Agrifood for providing the rainfall data, the technical staff of the Environmental and Life Sciences Programme, and Muhammad Abdul Hakeem bin Julaihi, Adrian Lee Rahman Suhaili, Siti Nisa Syahzanani Nafiah, Nur E'zzati Supri and Nurhazimah Ahmad for field and lab support.

**Conflicts of Interest:** The authors declare no conflict of interest.

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
