# Peer review of "Litterfall Production and Litter Decomposition Experiments: In Situ Datasets of Nutrient Fluxes in Two Bornean Lowland Rain Forests Associated with Acacia Invasion"

_data_

Round 1
Reviewer 1 Report
The authors describe the methodology and present data collected from a one-year study of litterfall production and leaf litter decomposition rates in two lowland Bornean forests affected by Acacia invasion. The manuscript nicely complements a paper published by the authors in the journal “forests” earlier this year.
The manuscript has a clear structure and is well written. Datasets such as the one presented in the paper are unique. The dataset is going to be particularly useful to those studying ecosystem-level effects of invasive species.
I have a few suggestions, hoping that these would improve the quality of the manuscript and accessibility of the dataset:
i) I suggest presenting a graphical summary of the variables of interest, displaying the variability present within the data. I understand that part of this information was already presented in the “forests” paper, but potential data users might be interested in getting a quick overview of the dataset without having to refer to the main paper.
ii) I would also encourage the authors to discuss the broad relevance of their dataset to scientists from different fields. Datasets such as the one presented in the paper are not only relevant to ecosystem ecologists, but potentially to other scientific communities, including earth system modellers, conservation biologists etc.
iii) Please consider changing the metadata file into a format that is more compliant with the FAIR data principles (this would also be in line with what is stated in the author guidelines for “Data” MDPI). The metadata file should be in a machine-readable format. (see https://www.go-fair.org/fair-principles/)
v) Please consider the possibility of using a unique identifier for the metadata. This would again be compliant with the FAIR data principles. Perhaps the dataset could be uploaded onto a repository such as zenodo or figshare.
vi) I would advise the authors to change the notation of the site coordinates (file GPS.csv). The sexagesimal degree notation ending with N or E makes it complicated to read the file into programming environments such as R and Python. Please consider the possibility of using a decimal degree notation.
Marco Girardello
Reviewer 2 Report
In the current study, a dataset on the impacts of invasive Acacia species on Bornean forests is provided. The followed methodology is acceptable and well communicated. My comments are provided below.
General comments
I have noticed many repetitions throughout the manuscript. Please avoid mentioning same things.
Specific comments
1. L18 I suggest avoiding any reference in the abstract section
2. L18-21 It is not clear what you mean, please revise
3. L70-71 This part has already been described.
L113-114 How randomness was ensured? Please, explain briefly.
